# Mouse Models in Meningioma Research: A Systematic Review

**DOI:** 10.3390/cancers13153712

**Published:** 2021-07-26

**Authors:** Julien Boetto, Matthieu Peyre, Michel Kalamarides

**Affiliations:** 1Department of Neurosurgery, Gui de Chauliac Hospital, Montpellier Universitary Hospital Center, 80 Avenue Augustin Fliche, 34090 Montpellier, France; j-boetto@chu-montpellier.fr; 2Institut du Cerveau et de la Moelle Épinière, INSERM U1127 CNRS UMR 7225, F-75013 Paris, France; matthieu.peyre@aphp.fr; 3Department of Neurosurgery, AP-HP, Hôpital Pitié-Salpêtrière, F-75013 Paris, France; 4Sorbonne Université, Université Pierre et Marie Curie Paris 06, F-75013 Paris, France

**Keywords:** meningioma, mouse model, xenograft, GEMM

## Abstract

**Simple Summary:**

Meningiomas are the most frequent primitive central nervous tumors in adults. Mouse models of cancer are used to study disease mechanisms and to establish preclinical drug testing. In this review, we describe all mouse models of meningiomas reported in the literature. This includes graft models wherein human meningioma cells are injected in nude mice, and genetically engineered mouse models. Taken together, these models have offered the possibility to study tumorigenesis mechanisms of initiation and progression and have provided useful tools for preclinical testing of a huge range of innovative drugs and therapeutic options. This review provides a systematic and comprehensive overview on how these different models can be used depending on the scientific questions to be answered.

**Abstract:**

Meningiomas are the most frequent primitive central nervous system tumors found in adults. Mouse models of cancer have been instrumental in understanding disease mechanisms and establishing preclinical drug testing. Various mouse models of meningioma have been developed over time, evolving in light of new discoveries in our comprehension of meningioma biology and with improvements in genetic engineering techniques. We reviewed all mouse models of meningioma described in the literature, including xenograft models (orthotopic or heterotopic) with human cell lines or patient derived tumors, and genetically engineered mouse models (GEMMs). Xenograft models provided useful tools for preclinical testing of a huge range of innovative drugs and therapeutic options, which are summarized in this review. GEMMs offer the possibility of mimicking human meningiomas at the histological, anatomical, and genetic level and have been invaluable in enabling tumorigenesis mechanisms, including initiation and progression, to be dissected. Currently, researchers have a range of different mouse models that can be used depending on the scientific question to be answered.

## 1. Introduction

Meningiomas are the most frequent primary central nervous system (CNS) tumors found in adults [1], representing approximately 30% of all intracranial neoplasms. Meningiomas can be divided into three prognostic histological subgroups according to the WHO classification: grade I (65–80%), grade II (20–35%, atypical), and grade III (<3%, anaplastic) with the three subgroups having very heterogeneous prognoses [2,3]. Surgery represents the standard treatment modality for benign meningiomas but is often insufficient to control grade II or grade III tumors, these tumors displaying aggressive biological features with high proliferation activity and/or infiltrative growth [3]. These cases often require adjuvant radiotherapy/radiosurgery in order to decrease the risk of recurrence. A large range of medical treatments have been proposed for refractory meningiomas, but they have shown only limited efficacy thus far [4]. 

The discovery of the mutational landscape of meningiomas raised new hopes for the possibility of using targeted therapies for refractory meningiomas. Genetic alterations of the *NF2* gene are found in 60% of sporadic meningiomas regardless of grade, suggesting an initial role in the meningeal tumorigenesis. About 40% of sporadic meningiomas are driven by non-*NF2* genetic mutations: these mutations essentially concern grade I meningiomas, where recent large-cohort sequencing studies have identified somatic coding alterations in *TRAF7*, *KLF4*, *POLR2A*, and members of the *PI3K* and Hedgehog signalling pathways [5]. 

As in the case in many other tumoral diseases, the establishment of preclinical meningioma animal models have attempted to mimic the genetic and biological alterations found in human in order to dissect the mechanisms of tumorigenesis (cell of origin, initiating events, mechanisms of progression…), and to assess the efficacy or toxicity of established or newly developed treatments. In this regard, several mouse meningioma models have been developed over decades, which have evolved in light of new discoveries in the comprehension of meningioma biology and with improvements of molecular biology and genetic technology to produce genetically modified mice. The multiplicity of the models and technologies to generate them as well as the high variability of their results can make it difficult to assess their real intrinsic value and their propensity to serve as a suitable model for a given objective. 

In this review, we summarize the different preclinical mouse models of meningiomas with special emphases on the technical aspects of their construction in order to highlight how they can facilitate the comprehension of meningioma tumorigenesis and how they represent essential tools for the evaluation of medical treatments. Our main objective was to critically analyze their specific characteristics in order to help researchers to choose the right model depending on their research goals and their financial and time resources. 

## 2. Materials and Methods

A systematic review of the literature identified from the PubMed database was performed in accordance with the PRISMA guidelines. A comprehensive search using keywords [“Meningioma” AND (“preclinical models” OR “mouse models”)] was conducted, including all studies from database inception until March 2021. Publications describing in vivo modelling of meningiomas in mice were included. Reviews, studies that did not publish a full manuscript, publications that were not in English and studies describing only in vitro results were excluded. 

## 3. Results

### 3.1. Articles Selection

A total of 142 articles were retrieved from the literature search. After the removal of duplicates and title/abstract screening for matching inclusion/exclusion criteria, 73 of these papers were then assessed for eligibility. Seventeen of these papers were excluded for the following reasons: review article, unclear methodology, in vitro study only, or subsequent retraction. Ultimately, 56 studies were included in the data analysis. The various approaches to the development of mouse meningioma models are summarized in Figure 1: this includes xenograft models (orthotopic or heterotopic) with human cell lines or patient derived tumors, and genetically engineered mouse models (GEMMs). Each of these models has advantages and disadvantages that are discussed in this review.

### 3.2. Meningioma Cell Lines

The vast majority of meningioma xenograft models employed immortalized cell lines obtained on human meningioma samples. The major obstacle in generating cell lines from benign meningiomas is the rapid occurrence of cell senescence when cells are cultured in vitro. Primary meningioma cell culture is restricted to early passages (senescence), due to low or no telomerase activity [6]. In the case of benign meningioma, the most common immortalization method employed was viral transduction of cells to generate the expression of the telomerase catalytic subunit (hTERT). Endogenous expression of hTERT is found in 30–50% of all benign meningiomas and nearly 100% of high-grade meningiomas [7]. Expression of hTERT in recurrent meningioma has also been observed [8]. Therefore, hTERT expression is a logical choice for manipulating tumor cell biology to permit continued cell growth in vitro. However, despite the careful characterizations described by the authors of those studies [9], it is difficult to assess what other aspects of the tumor cell biology may also have been altered, thus confounding the use of these cells as benign meningioma models. 

The principal meningioma cell lines are summarized in Table 1. The best-characterized line, which was derived from a benign WHO grade 1 meningioma, is the BenMen1 cell line [9]. This line exhibits typical cytological, immunocytochemical, ultrastructural and genetical features of meningiomas, including whorl formation, expression of epithelial membrane antigen, desmosomes, and interdigitating cell processes, as well as loss of chromosome 22q. Two other cell lines—SF-4433 [10] and Me3TSC [11]—have been developed with, in addition to the hTERT, co-expression of human papilloma genes E6/E7 and SV40 large T antigen, respectively, in order to achieve cell immortalization. However, these viral genes have not been associated with meningioma in vivo and such transformation by viral oncogenes could alter the growth signaling and behavior of these cells. 

On the other hand, some established cell lines derived from highly aggressive meningioma variants are available and have been used in the majority of xenograft models. The first cell line to be isolated was IOMM-Lee cells, derived from an anaplastic intraosseous meningioma, which showed extremely aggressive tumorigenicity in athymic nude mice which developed multiple pulmonary metastases [14]. While IOMM-Lee cells represent the most popular cell line used in preclinical studies, it has a complex karyotype, likely due to long-term culture, and suffers from a limited potential for generalized use in terms of studying disease-specific biology and novel treatments [17]. Moreover, it lacks the NF2 mutation, which is the main driver event of malignant meningiomas. In a recent study, a pair of cell clones characterized by either stable knockout of NF2 and loss of the NF2-protein merlin, or retained merlin protein, was generated using Crispr/Cas gene editing of the IOMM-Lee cell line [18]. The other popular cell lines used are described in Table 1 [12]. 

### 3.3. Mouse Xenograft Models

#### 3.3.1. Heterotopic Models

Heterotopic models, using primary cell culture or meningioma cell lines, were historically the first models to be developed [19]. Mouse xenograft meningioma models in the flank facilitate the follow-up of tumors growth and treatment efficacy and are low-cost. The classical technique is to subcutaneously inject 1 × 10^6^ cells, suspended in 0.5 mL of medium, into the flank of nude mice, and to then institute therapies 5 to 10 days after the injection [20]. This method has proven to be successful with tumor development found in about 60% of cases, particularly when mixing meningioma cells with Matrigel, a gelatinous basement membrane protein mixture secreted by Engelbreth-Holm-Swarm (EHS) mouse sarcoma cells, at the time of the subcutaneous injection [21]. Ragel et al. reported that meningioma cell lines or tumor-derived cultures that had multiple chromosomal abnormalities consistently induced tumors in the flank (whereas cell lines with normal karyotype did not). Flank tumors derived from cell lines exhibit histological, immunohistochemical and ultrastructural features that are consistent with meningiomas [20]. 

This approach enabled histologically confirmed meningiomas to be obtained and the monitoring of meningioma tumor cell growth both in primary cell cultures [22,23,24] and with meningioma cell lines [20,25,26]. The following treatments have been evaluated and showed an inhibition of tumor growth capacity in meningioma flank models: Verapamil [27], CREB-binding protein inhibitor ICG 001 [28], Irinotecan [25], Imatinib [29], fatty acid synthase inhibitors [30], farnesyl thiosalicylic acid [31], Siomycin 1 [32], Mifepristone [33], Pegvisomant [23], and Celecoxib [34]. However, none of these treatments have shown efficacy in treating meningiomas in humans. The ability of flank models to serve as real meningioma models (whose results can be transposed to humans with confidence) is not demonstrated, particularly since they lack the specific microenvironment of meningiomas (CSF, arachnoid, brain, and bone). In view of the progress made in limiting the morbidity of surgical procedures and progress in in vivo imaging, the orthotopic models should be preferred.

#### 3.3.2. Orthotopic Models

##### Injection Technique

Orthotopic models are generated by an intracranial injection of meningioma cells into 3–8-weeks-old immunocompromised mice. A wide variety of injection site, the type and number of cells injected, and injection volumes have been described and are listed in Table 2. 10^5^ to 10^6^ cells in a volume of 3 to 10 microliters is generally injected, though recently, xenografts with tumorosphere derived from malignant meningiomas were successfully implanted into the convexity, with only a very low number of implanted cells (50 × 10^3^) being necessary for tumor induction [35]. For this method, mice are anesthetized and their head is stabilized in small animal stereotaxic instrument. Subdural or convexity injections are performed through a burr hole drilled 2.5 mm lateral from the bregma, 1 mm deep in the skull. Skull base injections are generally performed through a burr hole drilled 1.5 mm anterior and 1.5 mm to the right of the bregma, with a needle slowly inserted downward about 5 mm [36]. An alternative option is the “post glenoid injection” technique: a 26-gauge needle tip is positioned on the right post-glenoid fossa (the rostral area of the opening of the external acoustic meatus). The implantation site, the lateral part of the foramen ovale, is accessed via a specific injection track [37].

##### Tumor Take and Meningioma Phenotype Results

The main results on tumor take are summarized in Table 2. The first orthotopic xenograft model of meningioma was established by Mc Cutcheon et al., using first passage primary cell cultures and the IOMM-Lee meningioma cell line [38]. Although the tumors maintained their relative phenotype of malignancy, they displayed several patterns of growth that would be unusual in human tumors, such as ventricular invasion and lepto-meningeal dissemination. Grafts using atypical and malignant meningiomas cell lines produced tumors in almost all immunocompromised mice that were injected regardless of the injection site (Table 2). Results using the implantation of immortalized benign meningioma cell lines are more heterogeneous, with tumor takes between 55 and 100%. Studies on patient-derived xenografts have shown more nuanced results with tumor rates ranging from 20% to 100%, even with anaplastic meningiomas, explaining why this technique is not considered to be completely reliable as a model for meningiomas [57]. 

##### Tumor Growth Monitoring and Mouse Imaging

Unlike heterotopic models, it is not possible to closely monitor tumor take and growth in orthotopic models in a simple manner. Two main techniques are currently used: bioluminescence-based methods and imaging using small-animal MRI. Baia et al. established intracranial xenografts using luciferase-expressing IOMM-Lee cells and used bioluminescence imaging (BLI) to quantify tumor growth. D-luciferin was injected intravenously in the mice prior to imaging to obtain bioluminescence, which was then detected through an ultra-sensitive camera. Using this method, the authors established the growth kinetics of meningiomas xenografts in vivo and demonstrated that the tumor volume was well-correlated with the mean tumor radiance [36]. BLI is a classical and relatively inexpensive way to monitor tumor growth and has been widely used in meningiomas models (through grafting of luciferase expressing cell lines [36,47,48,58], or crossing with luciferase reporter enabling bioluminescence imaging of Cre-loxP-dependant tumorigenesis, see below [59]). 

Other studies have demonstrated the feasibility of magnetic resonance imaging for monitoring meningioma formation, with tumors as small as 1–2 mm^3^ that were detectable and could be followed by sequenced imaging [60]. Moreover, dynamic contrast enhancement sequences showed their ability to reflect tumor perfusion and capillary permeability [59]. Major drawbacks of magnetic resonance imaging are its high costs and that it lacks ready availability. 

Finally, a recent study demonstrated the ability of radiolabeled somatostatin analogues (68Ga-DOTATATE) to detect meningiomas in subcutaneous xenografts of the CH-157MN meningioma cell line [61,62]. Its use in orthotopic models of meningiomas has not thus far been described.

##### Intraoperative Fluorescent Tumor Visualization

Recently, the possibility of the selective identification of meningioma cells in vitro and in vivo with fluorescent technique was described. FAM-TOC (5,6-Carboxyfluoresceine-Tyr3-Octreotide), a somatostatin receptor-labeled fluorescence dye, was able to be detected after incubation in vitro in various meningioma cell lines of all grades [63]. Moreover, meningioma cells grafted intracranially in vivo were able to be detected with fluorescence microscope or endoscope and enabled a fluorescent-guided resection [64]. This model represents a valuable experimental model for fluorescence meningioma surgery and in vivo imaging.

##### Limits of Xenografts Models

Xenograft models using established meningioma cell lines are reproducible with respect to tumor take and growth rate, but they require the use of an immunocompromised host, thus making it impossible to study interactions between tumor cells and the host immune system, an increasingly relevant field of study in meningiomas. In addition, a strong selective pressure is often observed during cell culture, raising the concern that cells used for experiments may no longer be representative of the original tumor. From this perspective, orthotopic models with cell lines represent strong models in order to screen for new therapeutic tools. 

On the other hand, models based on primary tumor grafts may more closely reflect the human pathology and serve as a personalized model at the scale of a unique patient, but often lack reproducibility and present wide variations in tumor take and growth rate. Taken together, these elements explain why xenograft models are not considered to be optimal for dissecting the biological events and mechanisms involved in meningioma tumorigenesis. The main objective of xenograft models is rather to provide tools for preclinical testing of innovative drugs, and a huge range of therapeutic options have been tested in those models (Table 2). Though the vast majority of these show efficacy in vitro and in vivo, results have not been confirmed in human studies [4]. This concern raises the question of the accuracy of xenograft models for prediction of the antitumoral effect of drugs, particularly in those models using immortalized malignant cell lines. Indeed, it is not completely clear how closely related the immortal meningioma cell lines are to the human tumors. However, due to their ease and speed of generation, they could represent a first step in the validation of new therapeutic options for meningioma treatment. 

### 3.4. Genetically Engineered Mouse Models (GEMM)

The molecular analysis of human meningiomas has been instrumental in the development of mouse models that closely resemble to their human counterpart. Ideally, a tumor model should be the closest transposition of its human counterpart regarding the histology, the anatomy, and the genetic driver events. Moreover, it should offer the possibility to control the tumor initiation from a temporal, spatial, and genetic perspective. GEMMs provide many of these features and allow to extensively manipulate genes [65]. In this section, the main genetic engineering technologies with their respective meningioma models (summarized in Table 3) are reviewed, as well as their main drawbacks. 

#### 3.4.1. Cre-loxP System

The Cre-loxP system is a site-specific recombinase technology that allows site-specific DNA modifications such as deletion, insertions, and translocations. This system was first used to better understand the role of allelic loss and/or mutation in the *NF2* gene at chromosome 22q (the main driver event in sporadic meningiomas). The *Nf2* homozygous germline null mouse model is lethal, and the hemizygous *Nf2* knock-out mouse (*Nf2*^+/−^) does not develop meningiomas [71], because the loss of the wild-type allele does not occur spontaneously in the murine meningeal cells, unlike in humans. Since no meningeal promotor was available at this time, the first models were based on direct meningeal cells targeting by intrathecal injection of an adenoviral vector (recombinant Cre Adenovirus) into the cerebrospinal fluid of *Nf2^loxP/loxP^* pups [66]. This technique had the advantage of targeting all meningeal cells, and thirty percent of mice developed a range of benign meningiomas subtypes that were histologically similar to the human tumors: transitional, meningothelial and fibroblastic (see Table 3 for details). In this first GEMM meningioma model, the initiating lesion associated with murine leptomeningeal tumorigenesis was defined as “meningothelial proliferation”, referring to microscopic lesions composed of meningothelial cells that represent early tumor formation. This model confirmed that biallelic *Nf2* inactivation was sufficient to induce meningioma and was a fundamental driver event in this context. 

Efforts were then made to generate models that could illustrate progression to higher grades, in an attempt to mirror the genetic events in human meningiomas. Alterations on chromosome 9p21 during meningioma progression have been found to induce losses of the tumor suppressor genes *CDKN2A* (*p16INK4a*), *p14ARF*, and *CDKN2B* (*p15INK4b*) [72,73]. Moreover, deletions of *CDKN2A*/*CDKN2B* are of poor prognostic factors in anaplastic grade III meningiomas [74]. The first attempt to induce the loss of *Cdkn2a,* via adding nullizygosity for *p16*Ink4a in *adCre*; *Nf2*^flox2/flox2^ mice, resulted in an increased rate of meningiomas development and meningothelial proliferation, but it did not modify the tumor grade (Table 3) [67]. Additional hemizygosity for p53 did not modify the frequency of meningioma nor malignancy, suggesting that, as in humans, *Nf2* and *p53* mutations did not synergize in promoting murine meningeal tumorigenesis [67]. Mechanisms of meningioma progression were confirmed using later models, wherein *Nf2* inactivation in synergy with homozygous or heterozygous *Cdkn2ab* deletions led to increased meningioma frequency and induced grade II and III meningiomas, thus representing a reliable atypical or anaplastic model that mimics the human pathology [59]. These models also offer the possibility of producing meningioma cell lines from mouse tumors and syngeneic orthotopic allografts to immunocompetent wildtype mice, which represent the closest models of sporadic malignant meningiomas [75]. 

Three cell lines (MGS1, MGS2, and MGS3) have been generated from grade I mouse meningiomas obtained in 4-month-old *AdCre*; *Nf2^flox2/flox2^*; *Inkab^−/−^* mice. Meningiomas were subsequently obtained in 100% of mice after subdural injection of these cells into immunocompetent FVB mice [59].

The discovery of the specific meningeal promotor PGDS has led to a second generation of GEMMs of meningiomas. The *prostaglandin-D2-synthase (PGDS)* gene was identified as a marker of meningeal cells in rats, mice, and humans [76,77,78]. PGDS is an enzyme responsible for the biosynthesis of prostaglandin D2 in the CNS. Several studies have reported that both human and mouse meningiomas exhibit intense PGDS immunoreactivity, suggesting that PGDS is a marker of normal and neoplastic meningeal cells [67,68,78]. PGDS appears in the WHO classification as a marker for the cell of origin in meningiomas. *PGDS* promotor-directed bi-allelic inactivation of *Nf2* led to the development of both meningothelial and fibroblastic meningiomas, whereas additional nullizygosity for *p16^Ink4a^* or *p53* did not increase the number or malignancy grade of meningiomas (Figure 2, Table 3) [59,68]. PGDS was also used to induce oncogenic somatic mutations that have been newly described in meningiomas, in meningeal cells. The *PGDS*Cre-*SmoM2* model has been generated, and the activating mutation of Smoothened from the early embryonic period resulted in formation of meningothelial meningioma at the skull base, as is seen in their human counterpart (Table 3) [70].

#### 3.4.2. RCAS-TVA System

The RCAS-TVA system is a popular gene delivery system to model human cancer [79]. It was used to demonstrate that overexpression of PDGF (platelet-derived growth factor), in meningiomas in arachnoïdal cells could induce meningiomas independently of *Nf2* mutation [69,80,81]. In this model, malignant progression could be induced by combining PDGF overexpression, *Nf2* mutation and additional loss of *Cdkn2ab* (Table 3) [69]. Unfortunately, PDGF-B overexpression in PGDStv-a expressing cells also induced gliomas of various histological grades, likely due to PGDS expression in oligodendrocytes. The establishment of these models led to the proposition of a specific classification for meningiomas in mice to be proposed, which differs slightly different from the WHO classification of their human counterparts (Table 4) [59].

#### 3.4.3. Limits of GEMMs

In addition to mirroring human meningioma biology, GEMMs of meningioma have contributed to a better understanding of the molecular mechanisms and spatio-temporal susceptibility to meningioma tumorigenesis. However, several drawbacks prevent GEMMs of meningiomas from being widely used. Time and financial costs for the generation and use of models can be prohibitive. They may require several crosses and the time to tumor growth can be very long (especially for grade I meningiomas). Unlike xenograft models, which are very reliable in term of tumor take rates, the tumor prevalence in GEMMs generally ranges from 30 to 80%, and the tumor growth rates and kinetics are unknown. Therefore, the presence of meningeal tumor presence must be generally confirmed through imaging, and these models are thus not systematically appropriate for preclinical testing. Additionally, these models sometimes result in non-meningeal tumors and consequently early death related to these (high grade gliomas, sarcomas, etc.).

It should be noted that the strategy of direct Adenovirus Cre or RCAS injection has the advantage of inducing a mutation in a small population of cells that are surrounded by, and must out-compete, their normal counterparts in vivo, accurately mimicking human cancers with the presence of wild-type competitor cells modulating the ability of mutant cells to induce disease.

### 3.5. Future Directions

#### 3.5.1. Next-Generation Mouse Modeling of Cancer with CRISPR/Cas9 Technology

Recently, the discovery of the clustered regularly interspaced short palindromic repeats (CRISPR) and CRISPR-associated proteins (Cas) has opened up the possibility of generating transgenic mice models at a lower cost [82,83,84]. Crispr technology enables the direct modification of genes in mice, or specific targeted modification of genes in cell lines that can be used in classical xenografts models. Prager et Al induced the knock-out of DUSP1 via this technique and could evaluate the biological impact of this event in vitro and in vivo after heterotopic grafting, as well as being able to study the therapeutic effects of a DUSP1/6 inhibitor [85].

#### 3.5.2. Extending GEMMs Models to Study Other Mutational Events and New Meningeal Promotors

New GEMMs are needed in order to explore targetable somatic mutations found in human meningiomas, such as *TRAF7*, *AKT1* or *PIK3CA*, where they will be essential for understanding the specific biological mechanisms involved and to provide accurate tools for preclinical drug evaluation. Moreover, new specific meningeal markers, that are potentially useful as meningeal promotors, must be discovered in order to better target specific subpopulations of meningeal cells. From this point of view, a better comprehension of meningeal embryology will undoubtedly provide new candidates and new opportunities in meningioma mouse modeling, as illustrated by the discovery of specific markers of primitive meningeal cell subpopulations through single-cell sequencing of the primitive meninges [86].

#### 3.5.3. Limitations

This review presents several limitations. First, we chose to focus on in vivo studies, and did not explore the field of “pure” in vitro studies. If a large number of drugs were tested in vitro, we believe that in vivo confirmation is mandatory before thinking about translation into human studies and that in vivo studies offer more robust biological conclusions. 

Second, GEMMs models are mainly described and published by a single team. The validity and usefulness of these models should be demonstrated by their use in a larger number of teams with similar results. 

## 4. Conclusions

The field of meningioma research has taken advantage of the development of several preclinical mouse models of meningioma (xenograft and transgenic models) to better understand the underlying biological mechanisms of meningioma tumorigenesis. Such models have also provided a means to test innovative potential therapies. Researchers now have a variety of available models that can be employed, depending on the specific research goal and on available financial resources. 

Our review demonstrates that the two main categories of meningioma mouse models have specific uses. On the one hand, orthotopic xenograft models offer a strong reliability in terms of tumor takes, at lower costs, and are therefore used for preclinical testing of new drugs or innovative treatments. However, the accuracy of these models for the prediction of the antitumoral effect of drugs, particularly in those models using immortalized malignant cell lines, is very questionable. This point is highlighted by the effectiveness of a large number of drugs on in vivo meningioma models that was not confirmed afterwards in human studies. 

On the other hand, GEMMs models strictly mirror the human biology and offer precious tools in order to dissect the meningioma tumorigenesis mechanisms, from both a spatial and a temporal manner. However, they are time-consuming and expensive, and present heterogeneous tumor take rates, making them inappropriate for large-scale preclinical drug testing studies. It should be noted that they also offer the possibility of generating mouse meningioma cell lines (both benign and malignant) and syngeneic orthotopic allografts to immunocompetent wildtype mice. This model gathers the advantages of both type of models, offers high tumor take rates, and represents the closest models of sporadic meningiomas. Future progress in both genetic and cell culture techniques will undoubtedly provide new opportunities for the development of innovative models. 

## Figures and Tables

**Figure 1 cancers-13-03712-f001:**
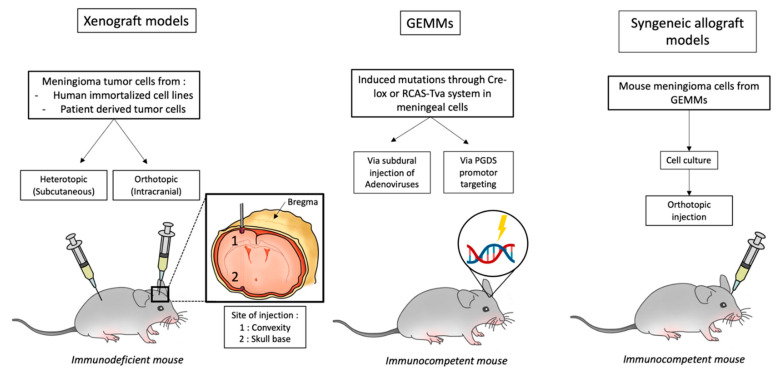
Illustration of different available mouse models of meningioma.

**Figure 2 cancers-13-03712-f002:**
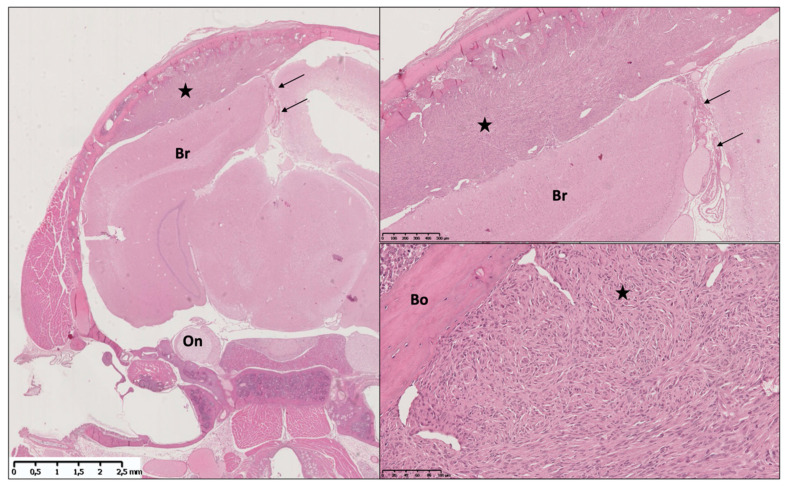
Illustration of a meningothelial meningioma in the AdCre; Nf2flox/flox; Ink4ab−/− model. Left image shows a coronal section of the whole cranium, with a massive right convexity meningioma (star) causing mass effect on the brain (Br). The lesion is centered at the Adenovirus injection site, a few millimeters lateral from the falx (arrows). Upper right image shows the meningioma at higher magnification; note that the brain cortex is not invaded in this case. Lower right image shows the tumoral architecture at higher magnification. Note the classical aspect of cellular whorl (star) and the close relationship with the skull convexity bone (Bo).

**Table 1 cancers-13-03712-t001:** Summary of principal meningioma cell lines.

Name	Phenotype	Immortalization	Genetics	Year	Ref.
HBL-52	Grade I, Meningothelial	-	*TRAF7* mutant	2008	[12]
Me3TSC	Grade I, Meningothelial	hTERT + SV40	-	2007	[11]
BenMen1	Grade I, Meningothelial	hTERT	22q loss *NF2* mutant	2005	[9]
SF4433	Grade I, Meningothelial	hTERT + HPV E6/E7	No 22q loss	2006	[10]
MENII-1	Grade II	hTERT + HPV E6/E7	22q loss	2008	[13]
IOMM-Lee	Grade III	-	No 22q loss	1990	[14]
CH 157 MN	Grade III	-	22q loss *NF2* mutant	1995	[15]
KT 21	Grade III	-	22q loss C-myc	1989	[16]

**Table 2 cancers-13-03712-t002:** Summary of meningioma orthotopic xenograft models.

Mouse Strain/Age of Injection (Weeks)	Injected Cell Types (WHO Grade)/Numbers/Volume (uL)	Site of Injection	Tumor Take (%)	Treatment	Clinical Results	Year, Reference
Athymic/6	IOMM-Lee (III); human tumor/10^6^/10	WM/floor of TF	85–100	-	-	2000, [38]
Athymic/6–8	IOMM-Lee (III)/10^6^/3	Floor of TF	100	Verotoxin	Inhibition of TG	2002, [26]
Athymic/6	BenMenI (I)	Convexity	100	-	-	2005 [9]
Athymic/6–8	IOMM-Lee (III)/5.10^5^	Brain	100	siRNA	Inhibition of TG	2006 [39]
Athymic/4	Me3TSC (I)-Me10T (I)/10^6^/5	Convexity (SDS)	100	-	-	2007, [11]
Athymic/3	CH-157-MN (III); IOMM-Lee (III)/10^4^–10^6^/3	Floor of TF/SDS convexity	90	Lb100 + RT	Increased survival compared to RT alone	2008, [17] 2018, [40]
Athymic/5–6	IOMM-Lee (III)/5.10^4^/0.5	Skull base	100	Temozolomide	Inhibition of TG Increased survival	2008, [36]
Athymic/5	KT21-DEP1 loss (III)	Convexity	100	-	-	2010, [41]
Athymic/5	PD grade I/10^6^/10	Convexity (prefrontal cortex)	90–100	Celecoxib	No effect	2012, [42,43]
Athymic/8–10	IOMM-Lee (III)/2.5 × 10^5^/5 uL	Convexity	100	Cliengitide + Radiotherapy Sorafenib Temsirolimus	Inhibition of TG	2013, [44,45,46]
Athymic/6–8	BenMen1(I)/10^6^/3	Skull base	100	Histone deacetylase inhibitor AR 42	Affect cell cycle progression Inhibition of TG	2013, [47]
NOD/SCID/gamma null mouse/8	IOMM-Lee (III)/5.10^4^/3	Skull Base (Pgi)	100	Peripheral blood mononuclear cells	Inhibition of TG	2013, [37]
Athymic/6–8	BenMenI (I)- KT21-MG1 (III)/10^6^/5	Skull base	100	Group 1 Pak inhibitor	Inhibition of TG	2014, [48]
Athymic/NA	PD grade I/10^6^/10	Convexity	100	-	-	2015, [49]
Athymic/5–6	Primary malignant meningioma *NF2*-mutant MN3/tumorosphere 50000/3–5	Convexity	100	Oncolytic HSV OS2966	Increased survival Increased Survival	2016, [35] 2019, [50]
Athymic/NA	CH-157MN (III)	Convexity	100	Hydroxyurea + verapamil	No effect	2016, [51]
Athymic/5–6	CH-157 MN (III)/5.10^4^/5 uL	Convexity/Skull base	55–80	-	-	2019, [52]
Athymic/6–8	IOMM-Lee (III)/10^4^	Skull base	100	-	-	2019, [53]
Athymic/6	KT21-MG1/50000/	Convexity	100	Mebendazole + /− Radiotherapy	Inhibition of TG Increased survival	2019, [54]
SCID mice/4–6	IOMM-Lee (III)/10^6^/3–10	Skull base	100	Ganoderic Acid DM	Inhibition of TG Increased survival	2019, [55]
SCID mice/4–6	IOMM-Lee (III)- BenMen1 (I)	Skull base	100	Palbiciblib + RT	Inhibition of TG Increased survival	2020, [56]

Abbreviations: PD: patient-derived; WM: white matter; TF: temporal fossa; SDS: subdural space; RT: radiotherapy; TG: tumor growth; Pgi: post-glenoid injection; SCID: severe combined immunodeficient; NOD: non-obese diabetic.

**Table 3 cancers-13-03712-t003:** Summary of meningioma GEMM models.

Construction	Genetics	Temporal Window of Activation	Phenotype (Grade)	Meningioma Prevalence	Year, Reference
*AdCre*; *Nf2^flox/flox^*	*Nf2* loss	PN2-PN3	M/F (I)	29% (TO) 19% (SD)	2002, [66]
*AdCre*; *Nf2**^flox/flox^*; *Ink4a*/**	*Nf2* loss + homozygous *P16^Ink4a^* mutation	PN2-PN3	M/F/T (I)	38% (TO) 36 % (SD)	2008, [67]
*AdCre*; *Nf2**^flox/flox^*; *Ink4ab**^−/−^*	*Nf2 + CDKN2AB* loss	PN2-PN3	66% (I) 31% (II) 3% (III)	72%	2013, [59]
*PGDS*Cre; *Nf2^flox:flox^*	*Nf2* loss in PGDS + cells	E12.5-PN2	M (I) F (I)	38% 38%	2011, [68]
*PGDS*Cre; *Nf2**^flox/flox^*; *p16**^ink4a/−^*	*Nf2* loss + *P16^ink4a^* mutation in PGDS + cells	E12.5-PN2	M (I) F (I)	50% 50%	2011, [68]
*PGDS*Cre; *Nf2**^flox/flox^*; *p53**^flox/flox^*	*Nf2* loss + p53 nullizygosity in PGDS + cells	E12.5-PN2	F (I)	43%	2011, [68]
*PGDStv-a*; PDGF-B	PDGF overexpression in PGDS + cells	E12.5-PN2	(I)	27%	2015, [69]
*PGDStv-a*; PDGF-B; *AdCre*; *Nf2^flox/flox^*	PDGF overexpression + *Nf2* loss in PGDS + cells	E12.5-PN7	Grade I (60%) Grade II (40%)	52%	2015, [69]
*PGDStv-a*; PDGF-B; *AdCre*; *Nf2^flox/flox^*; *Cdkn2ab**^−/−^*	PDGF Overexpression + *nf2* loss + *Cdkn2ab* loss	E12.5-PN7	Grade I (33%) Grade II (47%) Grade III (20%)	79%	2015, [69]
*PDGS*Cre; *SmoM2*	Activating mutation of *Smo* in PGDS + cells	E12.5-PN2	M (I)	21%	2017, [70]

Abbreviations: PN: post-natal day; M = meningothelial; F: fibroblastic; T: transitional; TO: transorbital; SD: subdural.

**Table 4 cancers-13-03712-t004:** GEMM classification for meningiomas [59].

Phenotype	Description
Meningothelial proliferation (early tumor formation)	Microscopic lesions composed of meningothelial cells
Grade I meningiomas	No mitotic figures, mild cytologic atypia, no brain invasion
Grade II meningiomas	One or two mitoses/HPF, true brain invasion
Grade III meningiomas	Marked cellular atypia (giant nuclei, pleomorphic nuclei, nuclei with marked chromatin clearing), brisk mitotic activity (three or more mitoses/HPF)

## Data Availability

Data from the systematic review are available on request from the corresponding author (M.K).

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
