# Peer review of "Mouse Models in Meningioma Research: A Systematic Review"

_cancers, 2021, doi:10.3390/cancers13153712_

Round 1

Reviewer 1 Report

The authors present a good overview and Review to mouse models for in vivo meningioma surgery. They should add following new publication also:

Fluorescence image-guided resection of intracranial meningioma: an experimental in vivo study on nude mice.

Linsler S, Müller SJ, Müller A, Senger S, Oertel JM.Linsler S, et al. Ann Anat. 2021 Sep;237:151752. doi: 10.1016/j.aanat.2021.151752. Epub 2021 Apr 30.   And discuss the fluorescence Imaging also.   Furthermore, the authors should present some translational informations (in vivo models and context to clinical work).     I suggest minor revivion before it mit be considered for publication.

Reviewer 2 Report

The introduction does not properly clarify what is the aim of this systematic review. If it is just a summary of the different preclinical mouse models of meningiomas and their uses, why the need for the review? In my opinion, the introduction should better expose what this review adds to the actual knowledge regarding mouse models of meningiomas. What is the objective? What is (are) the review question(s)?

The conclusion paragraph is also very short, is generic and lacks any relevant content. In my opinion, the conclusion section of a systematic review should provide some sort of interpretation of the findings in the review and also provide some discussion of issues arising from this findings. Potential limitations of the systematic review should also be listed here.
